# Studying an antiaromatic polycyclic hydrocarbon adsorbed on different surfaces

Zsolt Majzik [1], Niko Pavliček[1], Manuel Vilas-Varela [2], Dolores Pérez [2], Nikolaj Moll [1], Enrique Guitián[2], Gerhard Meyer[1], Diego Peña [2] & Leo Gross [1]

Antiaromatic and open-shell molecules are attractive because of their distinct electronic and magnetic behaviour. However, their increased reactivity creates a challenge for probing their properties. Here, we describe the on-surface and in-solution generation and characterisation of a highly reactive antiaromatic molecule: indeno[1,2-*b*]fluorene (**IF**). In solution, we generated **IF** by KI-induced dehalogenation of a dibromo-substituted precursor molecule and found that **IF** survives for minutes at ambient conditions. Using atom manipulation at low temperatures we generated **IF** on Cu(111) and on bilayer NaCl. On these surfaces, we characterised **IF** by bond-order analysis using non-contact atomic force microscopy with CO-functionalised tips and by orbital imaging using scanning tunnelling microscopy. We found that the closed-shell configuration and antiaromatic character predicted for gas-phase **IF** are preserved on the NaCl film. On Cu(111), we observed significant bond-order reorganisation within the *s*-indacene moiety because of chemisorption, highlighting the importance of molecule surface interactions on the $\pi$-electron distribution.

[1] IBM Research - Zurich, 8803 Rüschlikon, Switzerland. [2] Centro Singular de Investigación en Química Biolóxica e Materiais Moleculares (CiQUS), Departamento de Química Orgánica, Universidade de Santiago de Compostela, 15782 Santiago de Compostela, Spain. Correspondence and requests for materials should be addressed to Z.M. (email: maj@zurich.ibm.com) or to D.Pña. (email: diego.pena@usc.es)

Aromaticity is one of the most relevant, and intriguing, concepts in chemistry[1,2]. In 1931, Hückel suggested his well-known rule to explain the extra stability of planar monocyclic molecules that contain [4n + 2] π-electrons in a conjugated system[3,4]. With some limitations, the concept was later extended to polycyclic conjugated hydrocarbons (PCHs), based on the number of Clar sextets[5], or the presence of conjugated circuits with [4n + 2] π-electrons within a particular structure[6]. In 1967, Breslow introduced the term antiaromaticity as the inverse of aromaticity, in order to explain the destabilisation of molecules with [4n] π-electrons in a cyclic conjugated system[7]. Among the PCHs with antiaromatic character[8], indeno[1,2-b]fluorene (**IF**) is a remarkable example (Fig. 1)[9,10]. Compared with pentacene (**1**), the most prominent p-type organic semiconductor with five linearly fused six-membered rings and 22 π-electrons in its aromatic conjugate circuit ([4n + 2], $n = 5$), **IF** presents a 6-5-6-5-6 fused-ring motif and a formally antiaromatic 20 π-electron conjugate system ([4n] $n = 5$). The **IF** closed-shell configuration with a central eight π-electron *para*-quinodimethane core (in red, Fig. 1) and two Clar sextets (in blue) might be in resonance with the open-shell diradical configuration with three sextets[11,12]. As a result, unsubstituted **IF** is presumed to be an extremely reactive PCH and in fact has never been synthesised or even detected to date.

In contrast, in recent years large effort have been devoted to the preparation of substituted-**IF** derivatives that are stable enough to be isolated and studied in detail[13–17]. These antiaromatic PCHs have received special attention because of their distinct electron accepting, n-type semiconducting behaviour, based on their easy reduction to the corresponding aromatic dianions[13,18]. This feature, together with the narrow highest occupied molecular orbital (HOMO)–lowest unoccupied molecular orbital (LUMO) gap, make **IF** derivatives promising organic photoelectronic materials[19]. The high reactivity expected for unsubstituted **IF** implies a challenge for probing its structural and electrical properties. An interesting fact if that the parent structure of one of the most promising PCH cores for electronic devices has not yet been investigated[20].

Advances in atomic force microscopy (AFM), particularly in resolving[21] and modifying the structure of molecules at the atomic scale, have opened new routes in the chemistry of highly reactive compounds[22–30]. AFM with functionalised tips has been used to identify and characterise individual molecules[31–34]. One important aspect is the bond order, which can be resolved directly with AFM by comparing the apparent length and contrast of individual bonds within the molecule[31], which has already been applied for the on-surface characterisation of biphenylenes[35] and highly reactive molecules such as arynes[29] or diradicals[30].

Here we present the generation and characterisation of an antiaromatic PCH, i.e. **IF** comprising a π-system with 20 electrons. In-solution **IF** was generated by iodide-induced decomposition of dibromo-substituted precursor **2**, whereas on-surface **IF** was obtained by tip-induced dehydrogenation of the polycyclic hydrocarbon **3**. We found that **IF** survives for a few minutes in solution at ambient conditions. Using AFM and STM at low temperature and in ultra-high vacuum, we found that on bilayer NaCl on Cu(111) (denoted as 2 monolayers or 2 ML NaCl) **IF** preserves the closed-shell character predicted for the free molecule, whereas on Cu(111) the electronic configuration is significantly altered because of strong chemisorption to the surface.

## Results

**In-solution synthesis and lifetime of IF.** We explored the in-solution generation of **IF** by KI-induced decomposition of 6,12-dibromo-6,12-dihydro indeno[1,2-b]fluorene (**2**) (Fig. 1). It is

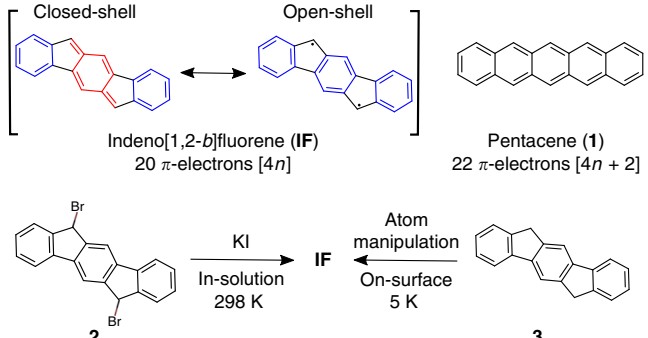

**Fig. 1** Molecular structures of indeno[1,2-b]fluorene (**IF**) and related compounds. The central eight π-electron *para*-quinodimethane core of **IF** is shown in red, the six π-electron Clar sextets are shown in blue

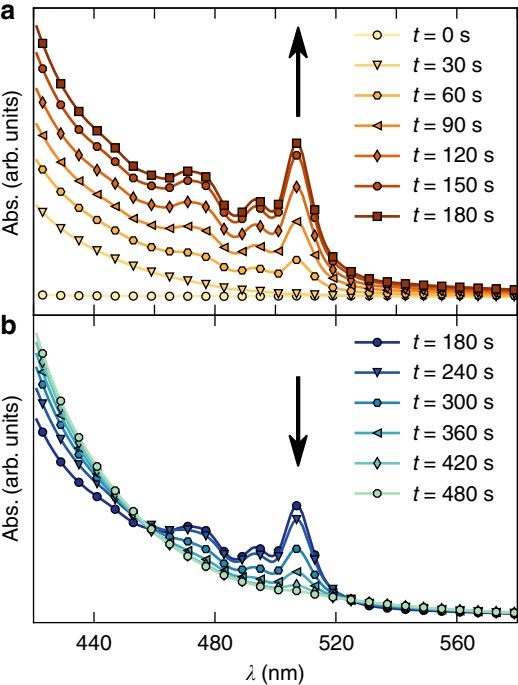

**Fig. 2** KI-induced dehalogenation of **2**. The reaction was monitored by UV spectroscopy measured in a 1:1 mixture of THF:CH$_3$CN at 20 °C. The first measurement ($t = 0$) was taken before the addition of KI. **a** Shows the time before the peak at 507 nm has reached its maximum and **b** after

well known that highly reactive *ortho*-quinodimethane can be generated by iodide-induced debromination from the corresponding precursors[36,37]. Based on these precedents, we synthesised compound **2** (see Supplementary Methods for details) to study the generation of **IF** at room temperature by treatment with KI in a mixture of THF:CH$_3$CN (1:1). The reaction was monitored by means of UV spectroscopy (Fig. 2). Before the KI treatment ($t = 0$), we did not observe any absorbance between 440 and 560 nm. However, immediately after adding KI, we observed a group of signals appearing between 460 and 520 nm ($\lambda_{max}$ ca. 507 nm), which are characteristic of the **IF** chromophore. Similar UV spectra were reported for 6,12-dimesityl-substituted **IF**, where there is a little electronic communication between the π-systems of the orthogonal mesityl groups and **IF** is very small[18]. The detected optical gap $E_{gap}$ and the position of the absorption peak $\lambda_{max}$ in our experiment are nearly identical with those of 6,12-dimesityl-substituted **IF** reported before (Table 1)[18]. The peak at 507 nm reaches its maximum intensity after 180 s of

**Table 1 Computational and optical data**

| Compound | Computational | | | Optical | | |
|---|---|---|---|---|---|---|
| | $E_{HOMO}$ | $E_{LUMO}$ | $E_{gap}$ | $\lambda_{max}$ | $\lambda_{edge}$ | $E_{gap}$ |
| **2** | −5.98 | −2.19 | 3.79 | 329 | 345 | 3.59 |
| **IF** | −5.42 | −3.17 | 2.25 | 507 | 534 | 2.33 |
| 6,12-Dimesityl-**IF**[18] | −5.32 | −3.15 | 2.17 | 516[18] | 541[18] | 2.29[18] |

Hybrid functional HSE (Heyd, Scuseria and Ernzerhof)[54, 55] with a mixing coefficient of 0.3 was applied for computational calculations. Wavelengths $\lambda_{max}$ and $\lambda_{edge}$ are in nm and energies $E_{HOMO}$, $E_{LUMO}$ and $E_{gap}$ are in eV

KI treatment; then it starts to vanish owing to the short lifetime of highly reactive **IF** (Supplementary Discussion).

**On-surface generation and characterisation of** IF. With the idea to increase the lifetime of **IF** and explore its aromaticity, we deposited compound **3** in ultra-high vacuum at low temperature on Cu(111) partly covered by bilayer NaCl islands. To dehydrogenate **3**, we followed a similar procedure as that for the dehydrogenation of triangulene precursors[38]. A Cu-terminated tip was positioned above the centre of a precursor molecule **3** at a tip height corresponding to an STM setpoint of $V = 0.1$ V and $I = 1$ pA. At the opened-feedback loop the tip was retracted by 4–8 Å to limit the tunnelling current to a few picoamperes at elevated biases, then the sample voltage $V$ was increased for 2 s. Typically, a sudden change in the tunnelling current occurred within two seconds at biases above 3.5 V, indicating a manipulation event[24,27,28,38,39]. The threshold for dehydrogenation (3.5 V) is consistent with previous experiments to remove a single H from doubly benzylic $CH_2$ groups[38] and with the dissociation energy of a C–H bond within the $CH_2$ group of fluorene (80 kJ/mol or 3.47 eV)[40]. On Cu(111), two hydrogens were removed sequentially, giving rise to two distinct steps in the current. On 2 ML NaCl, **IF** formed directly without the formation of a stable intermediate indicating concerted dehydrogenation (see Supplementary Fig. 6 and its discussion for more details about the on-surface generation of **IF**).

Figure 3 shows constant-height AFM images before and after dehydrogenation on both surfaces and also of the intermediate **3**′ after dissociation of only one H on Cu(111). AFM images were taken with a CO-functionalised tip at zero bias ($V = 0$). Height offsets $\Delta z$ are denoted with respect to an STM setpoint of $I = 1$ pA at $V = 0.1$ V above the respective substrate surface. A positive $\Delta z$ sign corresponds to an increase in the tip–sample separation. The structure of the precursor molecule adsorbed on Cu(111) was resolved using $\Delta z = -1.3$ Å. After complete dehydrogenation, molecular resolution was acquired at typically $\Delta z = -2.5$ Å, indicating a significantly reduced adsorption height. In contrast, on 2 ML NaCl we observed atomic resolution of **3** at $\Delta z = 1.7$ and of **IF** at 1.5 Å, suggesting only a minor change in the adsorption height due to dehydrogenation on NaCl. We quantified the change in the adsorption height with $\Delta f(z)$ spectroscopy using $z^*$, the tip height of which $\Delta f(z)$ is minimal, as a measure of the relative adsorption height (Fig. 4)[28]. The $z^*$ map acquired on Cu (2 ML NaCl) shows that **IF** adsorbs 0.94 Å (0.18 Å) closer to the surface than **3**. In addition, the $z^*$ maps indicate that **IF** adsorbs planarly on NaCl. On Cu(111) we observed a slight increase in the adsorption height at the outer benzene rings than at the molecular centre, similar to pentacene (**1**) adsorbed on Cu(111)[21,28].

In solution, the **IF** core can be easily transformed into [4n + 2] aromatic dication (18 π-electrons) or dianion (22 π-electrons) by oxidation or reduction reactions, respectively[13,18]. On 2 ML NaCl, the charge state of the adsorbed molecule is governed by

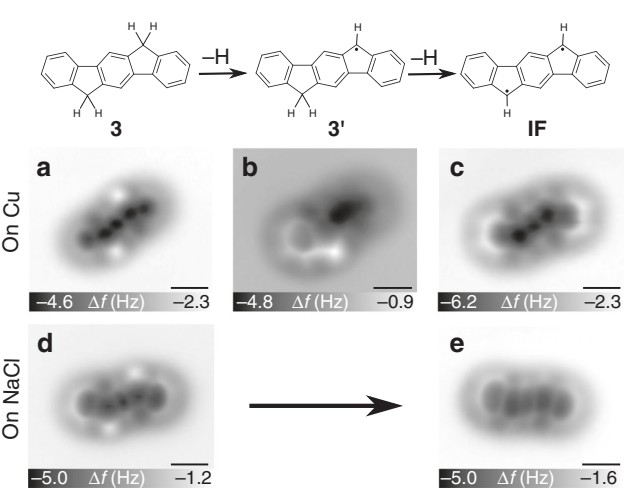

**Fig. 3** On-surface generation of indeno[1,2-*b*]fluorene (**IF**). 6,12-Dihydroindeno[1,2-*b*]fluorene (**3**) was dehydrogenated with bias pulses typically between 3.5 and 4.3 V. **a–e** Constant-height AFM images of the precursor molecule **3** (**a**, **d**), of radical **3**′ (**b**) and **IF** (**c**, **e**) measured with a CO tip at $V = 0$ on Cu(111) (**a–c**) and on 2 ML NaCl (**d**, **e**), respectively. Images were taken at tip height offsets **a** $\Delta z = -1.3$ Å, **b** $\Delta z = -2.0$ Å, **c** $\Delta z = -2.5$ Å, **d** $\Delta z = 1.7$ Å and **e** $\Delta z = 1.5$ Å with respect to the STM setpoint of $I = 1.0$ pA and $V = 0.1$ V above the respective substrate. Scale bars, 500 pm

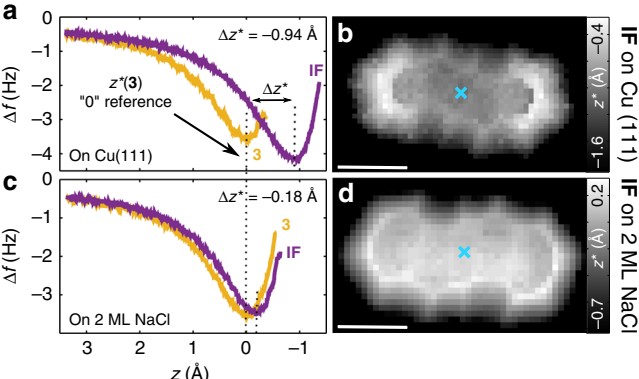

**Fig. 4** $\Delta f(z)$ spectroscopy. **a**, **c** $\Delta f(z)$ curves taken over the central benzene ring of **3** (yellow) and **IF** (purple) adsorbed on **a** Cu(111) and **c** 2 ML NaCl. The z scale is offset by $z^*$(**3**) for the respective surface to show **3** at $z = 0$ as reference. **b**, **d** $z^*$ maps of **IF** adsorbed on **b** Cu(111) and **d** 2 ML NaCl. At $z = -1.6$ Å on Cu(111) and $z = -0.7$ Å on 2 ML NaCl (black regions), the approach was aborted to avoid tip instabilities, and $z^*$ was not reached above the bare surfaces. Scale bars, 500 pm

the work function of the surface and by the electron affinity level and the ionisation potential of the molecule[41]. We observed no interface state electron scattering at **IF** on 2 ML NaCl, demonstrating that the molecule remains neutral and therefore antiaromatic on 2 ML NaCl[42].

To learn about the contributions of the open- and closed-shell resonant structures presented in Fig. 1, we probed the orbital configuration of **IF** on NaCl and carried out spin-polarised DFT calculations with a first-order perturbative correction ($G_0W_0$). For the calculations, we considered the molecule in its open-shell and closed-shell configurations (Methods). Their quasiparticle energies are shown in Fig. 5a. The zero of the energy scale has been adjusted to match the experimentally determined work function of 2 ML NaCl on Cu(111) ($\Phi = 4.0$ eV)[43]. Based on the level alignments, **IF** is predicted to remain uncharged in both cases

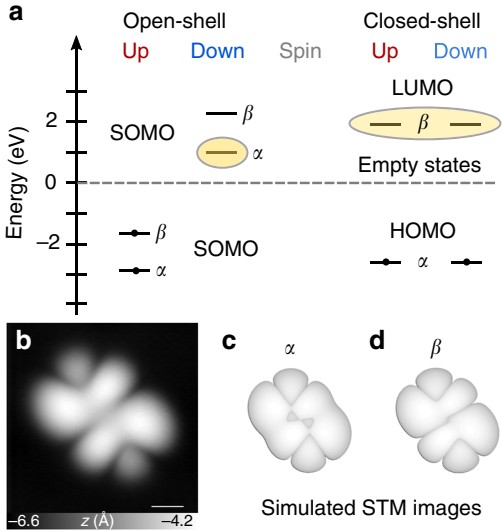

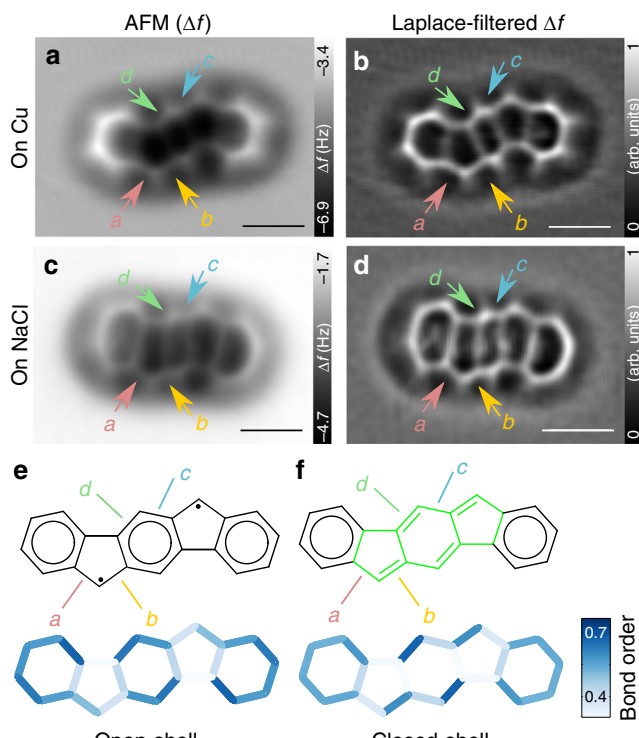

**Fig. 5** Spin-polarised DFT calculations. **a** Energy-level diagram obtained from DFT calculations carried out with the $G_0W_0$ approximation. Marked energy levels correspond to the lowest-laying unoccupied states. **b** Constant-current STM image of **IF** taken with a metal tip at the energy corresponding to the negative ion resonance (NIR) ($V = 1.0$ V and $I = 1.0$ pA). **c**, **d** Simulated STM images of orbitals $\alpha$ (**c**) and $\beta$ (**d**). Scale bar, 500 pm

**Fig. 6** Bond-order analyses. **a**, **c** Constant-height AFM images of **IF** taken with CO tip at $V = 0$ V on Cu(111) ($\Delta z = -2.1$ Å) (**a**) and on 2 ML NaCl ($\Delta z = 1.5$ Å) (**c**). **b**, **d** To emphasise the structure of the molecule, Laplace-filtered AFM images are also shown. **e**, **f** Bond orders of **IF** in its open-shell (**e**) and closed-shell (**f**) resonance structures. The s-indacene moiety is highlighted in green (**f**). On the bond-order scale, 0 refers to a single bond and 1 to a double-bond[45]. Bond orders were determined by using the bond lengths of relaxed geometries. Scale bars, 500 pm

considered. $\alpha$ and $\beta$ are the frontier molecular orbitals of **IF**. In the closed-shell configuration, $\alpha$ is fully occupied (closed) and is the highest occupied molecular orbital, whereas $\beta$ is the lowest unoccupied molecular orbital. Note that non-polarised energy minimisation also leads to the closed-shell configuration. In the open-shell configuration, $\alpha$ and $\beta$ are non-degenerate singly occupied molecular orbitals (SOMOs). We imaged the negative ion resonance (NIR) at positive sample bias, with its onset at $V = 1.0$ V (Fig. 5b). We were unable to image the positive ion resonance (PIR) because the molecule started to move at increased negative biases before the bias voltage reached the onset of the PIR. Because **IF** is neutral, the NIR is attributed to electron tunnelling into the lowest-lying empty state(s) of the free molecule[42]. In the closed-shell case, the NIR is expected to closely resemble the shape of the LUMO (orbital $\beta$), whereas in the open-shell case, the NIR is related to the empty spin-down channel of the $\alpha$ orbital (Fig. 5a). In Fig. 5c, d, simulated STM images are shown of orbitals $\alpha$ and $\beta$. We considered an extended s-like wave function for the tip to simulate the STM appearance of orbitals[38,44]. The experimentally observed image concurs with the simulated LUMO ($\beta$ orbital) STM image of the molecule in the closed-shell resonance structure, indicating that **IF** is in the closed-shell configuration.

In addition, we carried out a bond-order analysis by AFM, which can be used to investigate the contributions of open- and closed-shell resonant structures, as these resonant structures lead to qualitatively different bond-order relations in the s-indacene moiety. The bond-order analysis of **IF** on 2 ML NaCl also shows excellent agreement with the closed-shell configuration. Figure 6e and f shows bond orders derived from the relaxed geometries using Pauling's empirical model[29,45–47]. In experiment, we compare the brightness ($\Delta f$) and apparent lengths of bonds to deduce bond-order relations[28,29,47]. Bonds of greater bond order are imaged more brightly owing to greater repulsive forces and they appear shorter[47]. In the high-resolution AFM image on 2 ML NaCl shown in Fig. 6c, we compared bonds that show a very similar chemical environment, i.e. bond a with b and bond c with d, respectively. For **IF** on NaCl (Fig. 6c, d) the AFM

measurements indicate greater bond order of b than of a and greater bond order of d than of c, both in line with the closed-shell configuration of **IF**. In our calculations, the energy of the closed-shell configuration is by 0.92 eV lower than that of the open-shell for the gas-phase molecule. Our findings are in good agreement with the previously calculated low diradical character for **IF**[12].

On Cu(111), we were not able to probe the frontier molecular orbitals because of the strong electronic coupling between the molecule and the metal substrate[42]. However, bond-order analysis by AFM can be used to gain information about the resonance character of **IF**. AFM images appear less distorted on Cu(111) then on NaCl[48]. On Cu(111), bonds a and b appear with similar contrast and also their apparent bond lengths are comparable as well (Fig. 6a, b). On Cu(111), bond c, located in the central benzene ring of the s-indacene moiety, has a brighter appearance and a shorter apparent length than bond d. Thus AFM indicates that bonds a and b have even bond orders and c has a greater bond order than d. The bond-order relations found on Cu(111) match neither with the closed-shell nor with the open-shell configurations shown in Fig. 6e, f. As discussed above, the adsorption height of **IF** on Cu(111) decreases significantly with respect to its precursor **3**, indicating a strong molecule–surface interaction after dehydrogenation. Reduction of the adsorption height was shown for olympicene when the physisorbed state was changed into a chemisorbed configuration by transforming the molecule into a π-radical by atomic manipulation[28]. Similarly, we attribute the origin of the reduced

adsorption height and bond-order reorganisation to chemisorption.

In conclusion, we have shown a successful on-surface and in-solution generation and characterisation of highly reactive antiaromatic indeno[1,2-b]fluorene (**IF**). We generated **IF** by iodide-induced debromination in solution, whereas on surface we used tip-induced dehydrogenation. In solution, we found that **IF** survives for a few minutes even at ambient conditions. On surface, the molecule shows its antiaromatic, closed-shell configuration on 2 ML NaCl. This is in contrast to **IF** adsorbed on Cu(111), where bond-order analysis indicates significant deviations from the closed-shell configuration, demonstrating the importance of molecule–surface interactions on the $\pi$-electron distribution.

## Methods

**AFM/STM experiments**. Experiments were carried out using a home-built combined STM/AFM under ultrahigh vacuum conditions (below $10^{-10}$ mbar) at a temperature of 5 K. The bias voltage $V$ was applied to the sample. A qPlus sensor[49] (stiffness $k = 1800$ N/m, eigenfrequency $f_0 = 25$ kHz, quality factor $Q = 2 \times 10^5$) operated in frequency-modulation mode[50] was used to perform AFM measurements. A focused ion beam setup was used to cut and sharpen the PtIr tip. The oscillation amplitude was 0.5 Å. A Cu(111) single crystal was cleaned by several sputtering and annealing cycles. Ultrathin NaCl films were grown on Cu(111) by thermal evaporation of NaCl at a temperature of about 270 K. Low coverages of **3** and CO molecules were deposited while the sample temperature was kept below 10 K. CO tips were prepared by picking up a single CO molecule from NaCl[21].

**DFT calculations**. DFT calculations were performed using the FHI-AIMS code[51]. The geometry of the isolated molecule was optimised with the tight basis defaults. For structural relaxation, the Perdew–Burke–Ernzerhof exchange-correlation functional was applied[52] with vdW correction[53]. The convergence criterion for the total forces was $10^{-3}$ eV/Å, and for the total energy it was set to $10^{-5}$ eV. The closed-shell configuration was calculated by performing unrestricted spin-polarised energy minimisation or spin-unpolarised calculations. The open-shell configuration was considered by keeping a spin multiplicity of 3 for the total spin of the molecule during the spin-polarised total energy minimisation.

Hybrid functional Heyd, Scuseria and Ernzerho (HSE)[54,55] with a mixing coefficient of 0.3 was applied for the computational calculations of the HOMO–LUMO gaps presented in Table 1. The mixing coefficient was adjusted to have the best match between the optical gap and the calculated gap of 6,12-mesityl-**IF** and **2**. The convergence criterion for the total forces was $10^{-3}$ eV/Å, and for the total energy it was set to $10^{-5}$ eV.

**In-solution lifetime measurements**. UV/Vis spectra were recorded in a Jasco V-630 spectrophotometer. See Supplementary Methods for more details about in-solution experiments and the synthesis of the **IF** precursors **2** and **3**.

**Data availability**. All experimental and theoretical data presented here are available from the corresponding authors on reasonable request.

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

## Acknowledgements

We thank B. Schuler, S. Fatayer and R. Allenspach for discussions. We acknowledge financial support from the European Research Council Advanced Grant CEMAS (291194), the European Union project PAMS (610446) and the ERC Consolidator Grant AMSEL (682144) programs, the Agencia Estatal de Investigación (MAT2016-78293-C6-3-R and CTQ2016-78157-R), the Xunta de Galicia (Centro singular de investigación de Galicia accreditation 2016-2019, ED431G/09) and the European Regional Development Fund (ERDF).

## Author contributions

Z.M., N.P., G.M. and L.G. performed and analysed the AFM/STM measurements. Z.M., N.P. and N.M. carried out the DFT calculations. M.V.-V., D.Pér., E.G. and D.Peñ. synthesised precursor molecules and performed the in-solution experiments. All authors contributed to writing the paper.

## Additional information

**Competing interests:** The authors declare no competing interests.

