## [Peer Review File(PDF 273 kb) · Nature Communications]

Reviewers' comments:

Reviewer #1 (Remarks to the Author):

The manuscript by Majzik, Pena, et al. describes the preparation of the parent indeno[1,2-b]fluorene molecule in solution and on surfaces. There is tremendous interest in such hydrocarbon structures from both a fundamental standpoint (antiaromaticity, quinoidal vs. biradical bonding patterns) as well as an applied view (n-type materials for organic electronics). Given the significant interest, I believe this contribution to be both scientifically worthy and timely, such that inclusion in Nature Communications is certainly warranted. The paper is very well written, the results are clearly presented, and the citations of previous literature are entirely appropriate. Not surprisingly, the parent system is very unstable in solution and decomposes quickly. The STM/AFM images in Figures 3-6 are convincing and compelling that the authors have indeed generated what they claim. I do have a number of minor suggestions for the authors to consider for a revised manuscript:

1. I think the title of the manuscript could be modified a bit to be clearer as to what the authors want to demonstrate/accomplish:
"Open or closed shell: revealing the bonding character within an anti aromatic hydrocarbon".
2. On page 3 (and several other places in the manuscript), the authors claim that "we found that IF survives for a few seconds in solution..." Figure 2 shows that the concentration of IF peaks at ca. 180 seconds. Given that, I think the authors can feel free to claim that "we found that IF survives for a few MINUTES in solution..."
3. On page 4 the authors compare the UV-Vis data in Figure 2 with those of "substituted indenofluorenes with bathochromic shifts induced by the substitution with two ethynyl groups". I do not feel that this is the best comparison, given that the ethynyl groups are conjugated with the IF chromophore and thus shift the UV-Vis lambda max to ca. 600 nm. A better comparison would be the 6,12-dimesitylIF in reference 17, which has a lambda max of 515 nm, which is nearly identical 507 nm shown in Figure 2. The orthogonal mesityl groups contribute little to the conjugation within the IF skeleton, so this comparison seems to be the most valid.
4. In several places in the manuscript, "2 ML NaCl" is mentioned. Unless I missed it, I do not know what the "2 ML" means. Milliliters? Multilayers? This needs to be defined.
5. On page 7, the authors state they wish "to learn more about the aromaticity of IF". IF is 20 pi-electrons and thus antiaromatic, so a better word than "aromaticity" is "tropicity", as this word will allow the ring currents in the studied system to be either diatropic or paratropic.
6. On page 9, the word "aromaticity" is again not entirely fitting. Maybe say "The tropicity of IF

has a strong impact on the bond-order within the s-indacene moiety".

7. On page 11, similar concerns to the above:

(a) "...that IF survives for a few minutes even at ambient conditions".

(b) "...demonstrating the importance of the molecule-surface interaction on tropicity".

Once these suggestions are considered/incorporated, I strongly recommend acceptance in Nature Communications.

Reviewer #2 (Remarks to the Author):

Authors reported synthesis of a Hückel antiaromatic polycyclic hydrocarbon, indeno[1,2-b]fluorene, (IF) in solution and on two different surfaces. The synthesis is indeed a challenging task and state-of-the-art machinery employed for this work makes it interesting enough to be considered for publication in Nat. Commun. Yet, few points need careful consideration before I can recommend it for publication.

1- Characterization in solution is questionable. Authors claim that they identified IF by comparing the absorption spectrum of the molecule with similar structures reported in ANIE, 2011, 50, 11103 (reference S3). The comparison between these two spectra is very difficult in my opinion. The λ_{max} of supposed IF is found at 507 nm while that of similar compounds reported at 561-570 nm in reference S3; it is further stated in reference S3 that λ_{max} is rather insensitive to the substituents. Any potential peak below roughly 450 is masked by strong absorption of iodide and its complexes. Concerning the fact that fluorenyl anion, cation, and radical have relatively similar absorption spectra, I am afraid a single spectrum cannot unveil the secret of this reaction. Having a mixture of starting material (2) and KI may lead to rather complex reactions from radical pathway to the final product(7), not necessarily the expected concerted debromination.

-Have authors tried to remove oxygen from their reaction mixture and see how the life time and the final product (7) change?

-How about changing the temperature/solvent to optimize the condition for stabilizing potential IF for further analyses?

-State-of-the-art computations can help to define the difference between absorption spectra of potential intermediates (namely, IF and single radicals with different molecular charges) involved in the process.

2- Definition of the ground electronic state of the molecule needs more efforts. The authors on the basis of their NIR experiment concluded that since the first empty energy level of the molecule corresponds to their LUMO, obtained from DFT computations, the molecule must be closed shell. This could be true if the DFT computations were carried out in the presence of an external electric field that corresponds to the applied voltage in NIR experiment! A voltage of 1.0 V applied from a distance about 5 Å corresponds to nearly 0.004 au electric field. For a potential system with near-degenerate electronic states, such a field can change the electronic ground-state. Thus, there is no guarantee that the field-free LUMO corresponds to the LUMO in the presence of an external electric field.

3-DFT computations may terribly fail to identify the nature of ground-state for systems with (potential) multi-reference character; but MCSCF computations can help to define the ground-

state of IF. Authors may employ DMRG computations to unambiguously define the nature of ground-state. This comment hold true for reference 11 of the paper too.

4- Does the molecule remains totally planar on the NaCl surface? If yes, why the edges of the molecule are so bright in the AFM image? Does it mean that terminal bonds have the highest bond order? In addition AFM image on NaCl seems rather distorted compared with the image obtained on Cu surface. On Cu surface (the less distorted image) I can recognize that bonds (c) and (a) are brighter than (d) and (b). Doesn't it contradict the (potential) closed-shell nature of the molecule?

5-Huckel aromaticity/antiaromaticity must be really carefully interpreted in terms of stability. See ChemSelect, 2017, 2, 863 for further conceptual discussions.

Reviewer #3 (Remarks to the Author):

The paper reported the in-solution and on-surface synthesis of a parent anti-aromatic indenofluorene (IF). In solution, IF was generated by KI mediated debromination, and the transient species was determined by UV-vis absorption spectrum which is in accordance with the kinetically blocked IFs. However, due to the high reactivity, the in situ generated IF degraded quickly. On surface, the IF was generated by the well-developed method (manipulated by tip) and the bond order and electronic structures were analysed by nc-AFM and STM. The conclusion is in consistence with previous calculation that IF is a closed-shell anti-aromatic PCH.

While overall the work was carefully done and well presented, the impact of this work is limited because: (1) recently there are a number of similar works on on-surface synthesis of highly reactive species (e.g. arynes, triangulenes, acenes, etc, ...), and the method in this work is similar to those work; (2) it is well investigated that IF is a closed-shell anti-aromatic compound and the current results did not provide much new insight; (3) the tile containing "Open or Closed-shell" is misleading, there is actually not much discussion on this issue which on the other hand is quite clear based on previous studies.

Therefore, overall, I think the novelty and impact of this work do not justify a publication in Nature Communications.

Responses to Reviewers:

Responses:

Reviewer #1

The manuscript by Majzik, Pena, et al. describes the preparation of the parent indeno[1,2-b]fluorene molecule in solution and on surfaces. There is tremendous interest in such hydrocarbon structures from both a fundamental standpoint (antiaromaticity, quinoidal vs. biradical bonding patterns) as well as an applied view (n-type materials for organic electronics). Given the significant interest, I believe this contribution to be both scientifically worthy and timely, such that inclusion in Nature Communications is certainly warranted. The paper is very well written, the results are clearly presented, and the citations of previous literature are entirely appropriate. Not surprisingly, the parent system is very unstable in solution and decomposes quickly. The STM/AFM images in Figures 3-6 are convincing and compelling that the authors have indeed generated what they claim. I do have a number of minor suggestions for the authors to consider for a revised manuscript:

We thank the referee for the positive comments on the quality of our research and on its presentation within this manuscript.

1. I think the title of the manuscript could be modified a bit to be clearer as to what the authors want to demonstrate/accomplish:

"Open or closed shell: revealing the bonding character within an antiaromatic hydrocarbon".

We modified the title of the manuscript. It reads now as:

"Open or closed shell: studying an antiaromatic hydrocarbon adsorbed on different surfaces"

2. On page 3 (and several other places in the manuscript), the authors claim that "we found that IF survives for a few seconds in solution..." Figure 2 shows that the concentration of IF peaks at ca. 180 seconds. Given that, I think the authors can feel free to claim that "we found that IF survives for a few MINUTES in solution..."

We adjusted accordingly.

3. On page 4 the authors compare the UV-Vis data in Figure 2 with those of "substituted indenofluorenes with bathochromic shifts induced by the substitution with two ethynyl groups". I do not feel that this is the best comparison, given that the ethynyl groups are conjugated with the IF chromophore and thus shift the UV-Vis lambda max to ca. 600 nm. A better comparison would be the

6,12-dimesitylIF in reference 17, which has a lambda max of 515 nm, which is nearly identical 507 nm shown in Figure 2. The orthogonal mesityl groups contribute little to the conjugation within the IF skeleton, so this comparison seems to be the most valid.

We thank for the suggestion. We modified the manuscript to compare our in-solution results with 6,12-dimesityl-substituted IF, as suggested. In addition, we performed further DFT calculations utilizing the FHI-AIMS code and hybrid functional (HSE). We found a good agreement between the calculated gap of IF and the optical gap determined from UV/VIS spectroscopy.

4. In several places in the manuscript, "2 ML NaCl" is mentioned. Unless I missed it, I do not know what the "2 ML" means. Milliliters? Multilayers? This needs to be defined.

ML corresponds to monolayer(s). We studied the orbital structure of IF on bilayer NaCl with the thickness corresponding to 2 ML of NaCl. We clarified explicitly the meaning of ML in the manuscript.

5. On page 7, the authors state they wish "to learn more about the aromaticity of IF". IF is 20 pi-electrons and thus antiaromatic, so a better word than "aromaticity" is "tropicity", as this word will allow the ring currents in the studied system to be either diatropic or paratropic.

The referee is correct, the use of the term aromaticity is not appropriate here, as we determined already the number of electrons in the π -system. What we want to study in more detail is the contribution of the resonance structures (shown in Fig 1.) *i.e.* the open-/ closed shell character of IF (related to the tropicity), via analyzing the bond-order relations and orbital imaging. Our experiments are not affected directly by the ring currents (as NMR), therefore we would avoid using the term tropicity to avoid any confusions.

We made the mentioned text clearer: *"To learn about the contributions of open- and closed-shell resonant structures presented in Fig. 1, we probed the orbital configuration of IF on NaCl and we carried out spin-polarised DFT calculations with a first-order perturbative correction (GOWO).."*

6. On page 9, the word "aromaticity" is again not entirely fitting. Maybe say "The tropicity of IF has a strong impact on the bond-order within the s-indacene moiety".

Adjusted to: *"In addition, we carried out bond-order analysis by AFM, which can be used to investigate the contributions of open- and closed shell resonant structures, as these resonant structures lead to qualitatively different bond-order relations in the s-indacene moiety"*

7. On page 11, similar concerns to the above:

(a) "...that IF survives for a few minutes even at ambient conditions".

We adjusted.

(b) "...demonstrating the importance of the molecule-surface interaction on tropicity".

It now reads as follows: *"...demonstrating the importance of molecule-surface interactions on the π -electron distribution."*

Reviewer #2

Authors reported synthesis of a Hückel antiaromatic polycyclic hydrocarbon, indeno[1,2-b]fluorene, (IF) in solution and on two different surfaces. The synthesis is indeed a challenging task and state-of-the-art machinery employed for this work makes it interesting enough to be considered for publication in Nat. Commun. Yet, few points need careful consideration before I can recommend it for publication.

We thank the referee for the very detailed comments on our manuscript and for considering our work for publication in Nature Communications.

1- Characterization in solution is questionable. Authors claim that they identified IF by comparing the absorption spectrum of the molecule with similar structures reported in ANIE, 2011, 50, 11103 (reference S3). The comparison between these two spectra is very difficult in my opinion. The λ_{\max} of supposed IF is found at 507 nm while that of similar compounds reported at 561-570 nm in reference S3; it is further stated in reference S3 that λ_{\max} is rather insensitive to the substituents. Any potential peak below roughly 450 is masked by strong absorption of iodide and its complexes.

As mentioned in comment 3 of reviewer 1, we have modified this part to compare our experimental results with 6,12-dimesitylindeno[1,2-b]fluorene since orthogonal mesityl substituents have little electronic influence on the IF core (see ref. 18). The excellent agreement between the relevant peaks in our UV spectrum and the one shown in Ref. 18, supports further our idea that we have detected IF in solution.

Concerning the fact that fluorenyl anion, cation, and radical have relatively similar absorption spectra, I am afraid a single spectrum cannot unveil the secret of this reaction. Having a mixture of starting material (2) and KI may lead to rather complex reactions from radical pathway to the final product(7), not necessarily the expected concerted debromination.

-Have authors tried to remove oxygen from their reaction mixture and see how the life time and the final product (7) change?

The initial UV experiments were already performed with deoxygenated solvents to increase the lifetime of IF in solution. However, the experimental setups do not allow us to quantify the amount of oxygen present during the reaction.

-How about changing the temperature/solvent to optimize the condition for stabilizing potential IF for further analyses?

We thank the referee for the suggestion. We performed the reaction at different temperatures (5, 10 and 20 °C) and we found that the peak at 507 nm persists longer at lower temperature, suggesting that the lifetime of IF increases significantly as the temperature decreases (see Figure S2). In addition, we have found that the use of a THF/MeCN mixture of solvents is important to achieve the IF generation.

-State-of-the-art computations can help to define the difference between absorption spectra of potential intermediates (namely, IF and single radicals with different molecular charges) involved in the process.

We performed DFT calculations using a hybrid functional method (HSE). Among potential intermediates (IF and single radicals: neutral, anion and cation) shown in Table S1, the calculated gap of IF shows the best agreement with the optically determined gap of 2.33 eV, supporting that we have detected IF in

solution. To benchmark our parameter set for hybrid functional calculations, we calculated the gap of 6,12-dibromo-6,12-dihydroindeno[1,2-b]fluorene and that of 6,12-dimesitylindeno[1,2-b]fluorene (measured in Ref. 18). In both cases we found an excellent agreement between experiment and theory (see Table I in the main text and Table S1 in supplement).

2- Definition of the ground electronic state of the molecule needs more efforts. The authors on the basis of their NIR experiment concluded that since the first empty energy level of the molecule corresponds to their LUMO, obtained from DFT computations, the molecule must be closed shell. This could be true if the DFT computations were carried out in the presence of an external electric field that corresponds to the applied voltage in NIR experiment! A voltage of 1.0 V applied from a distance about 5 Å corresponds to nearly 0.004 au electric field. For a potential system with near-degenerate electronic states, such a field can change the electronic ground-state. Thus, there is no guarantee that the field-free LUMO corresponds to the LUMO in the presence of an external electric field.

We are not aware of any STM measurements where a reversal of the orbital order due to the applied field was observed. (It would be a great and unexpected result if that was the case and we would be very happy if we observe that, but unfortunately, we have no indication for it). But to take into account the skepticism of the referee we state that the STM results only indicate the closed shell character on NaCl.

p. (9) “... indicating that **IF** is in the closed-shell configuration.”

Our bond-order analysis on NaCl, measured at $V = 0$ V, independently corroborates and confirms the closed shell character of **IF** on NaCl.

3-DFT computations may terribly fail to identify the nature of ground-state for systems with (potential) multi-reference character; but MCSCF computations can help to define the ground-state of IF. Authors may employ DMRG computations to unambiguously define the nature of ground-state. This comment hold true for reference 11 of the paper too.

Our spin-polarized energy minimization indicates that the closed-shell configuration has lower energy than the open-shell configuration. Our experimental findings are in excellent agreement with DFT calculations, therefore we do not feel that further calculations are needed to better define the ground state of **IF** at this point.

4- Does the molecule remains totally planar on the NaCl surface? If yes, why the edges of the molecule are so bright in the AFM image? Does it mean that terminal bonds have the highest bond order? In addition AFM image on NaCl seems rather distorted compared with the image obtained on Cu surface. On Cu surface (the less distorted image) I can recognize that bonds (c) and (a) are brighter than (d) and (b). Doesn't it contradict the (potential) closed-shell nature of the molecule?

There are several points to be discussed. First, it is important to state, that both increased adsorption height and increased bond order result in increased frequency shifts (increased brightness). Therefore, these effects are difficult to deconvolve and bond-order should only be compared for bonds with equal adsorption height. Moreover, the non-even background due to van der Waals and electrostatic forces makes comparison of bond-orders challenging at the edges of the molecules, where the background due to these forces is not planar, see Gross et al. Science, 337, 1326 2012, ref [49]. For this reason, we only

compare bonds a with b and c with d. The compared bonds have very similar chemical environment and very similar height.

We show the measurement of the adsorption geometry of **IF** on NaCl and Cu(111) in Fig. 4 using the method described by Schuler et al. PRL 111, 106103 (2013) [ref 28]. On Cu(111), the ends of the molecule bend away from the surface, whereas on NaCl the molecule remains flat (within about 0.1 Å). This can be observed in Fig. 4b) on Cu (ends are bright) and in Fig. 4d on NaCl (whole circumference of molecule of approx. equal brightness. Note that in this plot the value z^* is plotted and thus the brightness values correspond to heights, which can be interpreted as adsorption heights (see Schuler et al. PRL 111, 106103 (2013) [ref 28]). The central bonds, which are part of two rings, are slightly less bright than the circumference for both surfaces, explained by the nonplanar background from van der Waals and electrostatic forces.

In agreement, in the const. height AFM images shown in Fig. 3, and Fig. 6 the molecule shows ends with significantly increased brightness on Cu(111) (Fig. 3c, Fig. 6a) whereas on NaCl (Fig. 3e, Fig. 6c) the whole circumference of the molecule shows less variations in brightness, with small modulations explained by bond-order differences and different background forces. The slightly brighter appearance of the ends in the const. height AFM images on NaCl results primarily from the differences in background forces. These differences are compensated in the z^* measurement (Fig. 4d), where we measure adsorption heights and observe no brighter appearance of the ends of the molecule compared to the other C-C bonds of the molecule. In addition, we calculated the adsorption geometry of **IF** adsorbed on Cu and on NaCl. In good agreement with our experimental findings, our calculations indicate that on NaCl **IF** adsorbs planar. Whereas, on Cu the molecule has a distorted geometry due to binding to the substrate (see DFT - adsorption geometries in supplement).

The observation of the referee that on NaCl the distortions are greater than on Cu(111) is correct. This effect was described in detail in Neu et al. PRB 89, 205407, 2014 [ref. 50]. It is explained by the greater attractive vertical force that the CO at the tip experiences above Cu(111) compared to NaCl. This greater (restoring) force leads to an increased lateral spring constant on Cu and thus smaller distortions on Cu compared to NaCl. The bond-order analysis, which is based on qualitative comparison of bonds can be performed independently on each surface.

By bond-order analysis we find that on NaCl the molecule is in the closed shell configuration. On Cu(111) we do not claim that the molecule is closed shell. It is one of the findings of the paper that on Cu **IF** is not in the closed shell configuration, in contrast to NaCl. On Cu we observe an altered bond-order sequence matching neither with closed-shell nor with the open-shell configurations, explained by chemisorption of the molecule. The observation of the referee that on Cu(111) c is brighter than d (in opposition to NaCl) is an important observation in this context. Comparing bond a with b we observe similar length and brightness on Cu(111), whereas on NaCl we observe that b is brighter and appears shorter than a.

We also inserted more often which particular surface we are discussing (NaCl or Cu(111)) to avoid confusion and to make each sentence independently readable and self-contained, without the need of reading the previous sentences to understand which substrate is discussed.

5-Huckel aromaticity/antiaromaticity must be really carefully interpreted in terms of stability. See ChemSelect, 2017, 2, 863 for further conceptual discussions.

We agree with reviewer 2 there is not always a direct relation between antiaromaticity and instability. Therefore we have included the suggested reference in the introduction of the manuscript as reference 4.

Reviewer #3

The paper reported the in-solution and on-surface synthesis of a parent anti-aromatic indenofluorene (IF). In solution, IF was generated by KI mediated debromination, and the transient species was determined by UV-vis absorption spectrum which is in accordance with the kinetically blocked IFs. However, due to the high reactivity, the in situ generated IF degraded quickly. On surface, the IF was generated by the well-developed method (manipulated by tip) and the bond order and electronic structures were analysed by nc-AFM and STM. The conclusion is in consistence with previous calculation that IF is a closed-shell anti-aromatic PCH.

While overall the work was carefully done and well presented, the impact of this work is limited because: (1) recently there are a number of similar works on on-surface synthesis of highly reactive species (e.g. arynes, triangulenes, acenes, etc, ...), and the method in this work is similar to those work;

We thank the referee for the positive comments on the quality of our research and on its presentation within this manuscript. We would like to stress that **IF** is not simply another highly reactive molecule. As it is commented by referee 1 *"There is tremendous interest in such hydrocarbon structures from both a fundamental standpoint (antiaromaticity, quinoidal vs. biradical bonding patterns) as well as an applied view (n-type materials for organic electronics)"*.

(2) It is well investigated that IF is a closed-shell anti-aromatic compound and the current results did not provide much new insight; the tile containing "Open or Closed-shell" is misleading, there is actually not much discussion on this issue which on the other hand is quite clear based on previous studies. Therefore, overall, I think the novelty and impact of this work do not justify a publication in Nature Communications.

We agree that the method for generation of **IF** on surface is similar to previous works. However, this is not the main point of our paper. It is, as the title states the investigation of this antiaromatic molecule and the characterization of its open or closed-shell character on different surfaces. So far only one study with AFM has investigated a formally antiaromatic biphenylene on surface, Kawai et al., ref [35]. However, biphenylenes are stable molecules which behave as two aromatic cores connected by two single bonds within a four-membered ring. In addition, they synthesized it by thermal annealing and only on a metal support. Here we demonstrated, that on metal we can expect strong chemisorption leading to a significant change in the bond-order relations and the open/ closed shell character of the free molecule. In contrast, on NaCl we observe the molecule in its closed-shell configuration which is the one expected for the free molecule. Our findings show that the open-/ closed shell character **on surfaces** is not quite so clear. We modified the title to highlight the importance of the surfaces.

The main part of the paper deals with investigating the open-/closed-shell character on these different surfaces. We do that by orbital imaging with STM, DFT calculations, and bond-order analysis by AFM. To make clear how these characterization methods relate to the open-/closed shell character of the molecule we wrote:

(p.8) *“To learn about the contributions of open- and closed-shell resonant structures presented in Fig. 1, we probed the orbital configuration of IF on NaCl and we carried out spin-polarised DFT calculations with a first-order perturbative correction (GoW₀).”*

(p.9) *“In addition, we carried out bond-order analysis by AFM, which can also be used to investigate the contributions of open- and closed shell resonant structures, as these resonant structures lead to qualitatively different bond-order relations in the s-indacene moiety.”*

We think the current title is justified, because we investigate the open/ closed shell character of **IF** using bond-order analysis by AFM on two different surfaces and orbital imaging by STM.

Reviewers' Comments:

Reviewer #1 (Remarks to the Author):

The revised manuscript by Majzik, Peña, et al. describes the preparation of the parent indeno[1,2-b]fluorene molecule in solution and on NaCl and Cu(111) surfaces. I first reviewed this manuscript this past July. As I stated then, there is tremendous interest in such hydrocarbon structures from both a fundamental standpoint (antiaromaticity, quinoidal vs. biradical bonding patterns) as well as an applied view (n-type materials for organic electronics), despite the somewhat odd comments of referee 3 about this not being important. Given the significant interest in IFs, I believe this contribution to be both scientifically worthy and timely, such that inclusion in Nature Communications is certainly warranted, especially now at the revision stage. The paper is very well written, the results are clearly presented, the citations of previous literature are entirely appropriate, and the attention to include/adapt the revision requests of referees 1 and 2. Not surprisingly, the parent system is very unstable in solution and decomposes quickly. The STM/AFM images in Figures 3-6 are convincing and compelling that the authors have indeed generated what they claim. I have a very few minor suggestions for the authors to consider for a re-revised manuscript:

1. I think the title of the manuscript could still be modified a bit more to be clearer as to what the authors want to demonstrate/accomplish:
“Open shell diradical or closed shell antiaromatic: studying a polycyclic hydrocarbon adsorbed on different surfaces”.
2. On page 3, why is it “unexpectedly” that the IF survives for a few minutes? Given the high degree of reactivity observed for many antiaromatic molecules, this short lifetime is exactly what I would have expected.
3. On page 4, the “group of signals” should be given as a range. How about the following: “...we observed a group of signals appearing between 460-520 nm (λ -max ca. 507 nm), which are characteristic...”
4. Reference 20 needs to be updated as it has page numbers now.

I will leave it to reviewer 2 to comment further as to the suitability/interpretation of the surface images. That said, it is clear to this reviewer that the authors have made an excellent effort to incorporate the changes requested by reviewers 1 and 2, and in doing so have improved an already strong manuscript. Once my minor suggestions are considered/incorporated, I strongly recommend acceptance in Nature Communications.

Reviewer #2 (Remarks to the Author):

I am pleased by detailed response to my comments by authors and gladly recommend the manuscript for publication. I just want to add one point for future studies, of course not in this case. As I mentioned in my previous comments, an electric field must potentially change the order of MOs. Just think what happens in a field effect transistor. This can be a potential subject for future studies, if authors find the right candidate.

Reviewer #3 (Remarks to the Author):

The authors indeed improved the quality of their manuscript according to the suggestion from reviewers 1 and 2. However, I still believe that this work does not represent a conceptual breakthrough neither in the area of diradical chemistry nor in the surface chemistry. It is an increase and does not provide much new insight. I leave the editor to decide based on their standard.

Responses to Reviewers:

Responses:

Reviewer #1

The revised manuscript by Majzik, Peña, et al. describes the preparation of the parent indeno[1,2-b]fluorene molecule in solution and on NaCl and Cu(111) surfaces. I first reviewed this manuscript this past July. As I stated then, there is tremendous interest in such hydrocarbon structures from both a fundamental standpoint (antiaromaticity, quinoidal vs. biradical bonding patterns) as well as an applied view (n-type materials for organic electronics), despite the somewhat odd comments of referee 3 about this not being important. Given the significant interest in IFs, I believe this contribution to be both scientifically worthy and timely, such that inclusion in Nature Communications is certainly warranted, especially now at the revision stage. The paper is very well written, the results are clearly presented, the citations of previous literature are entirely appropriate, and the attention to include/adapt the revision requests of referees 1 and 2. Not surprisingly, the parent system is very unstable in solution and decomposes quickly. The STM/AFM images in Figures 3-6 are convincing and compelling that the authors have indeed generated what they claim. I have a very few minor suggestions for the authors to consider for a re-revised manuscript:

1. I think the title of the manuscript could still be modified a bit more to be clearer as to what the authors want to demonstrate/accomplish:

“Open shell diradical or closed shell antiaromatic: studying a polycyclic hydrocarbon adsorbed on different surfaces”.

We thank the referee for the suggestion, unfortunately the title cannot contain a punctuation. We shortened it and now reads as follows: Studying an antiaromatic polycyclic hydrocarbon adsorbed on different surfaces.

2. On page 3, why is it “unexpectedly” that the IF survives for a few minutes? Given the high degree of reactivity observed for many antiaromatic molecules, this short lifetime is exactly what I would have expected.

We removed the word “unexpectedly”.

3. On page 4, the “group of signals” should be given as a range. How about the following: “...we observed a group of signals appearing between 460-520 nm (λ -max ca. 507 nm), which are characteristic...”

We adjusted according to the referee’s suggestion.

4. Reference 20 needs to be updated as it has page numbers now.

We updated the reference. Thank you.

I will leave it to reviewer 2 to comment further as to the suitability/interpretation of the surface images. That said, it is clear to this reviewer that the authors have made an excellent effort to

incorporate the changes requested by reviewers 1 and 2, and in doing so have improved an already strong manuscript. Once my minor suggestions are considered/incorporated, I strongly recommend acceptance in Nature Communications.

Reviewer #2

I am pleased by detailed response to my comments by authors and gladly recommend the manuscript for publication. I just want to add one point for future studies, of course not in this case. As I mentioned in my previous comments, an electric field must potentially change the order of MOs. Just think what happens in a field effect transistor. This can be a potential subject for future studies, if authors find the right candidate.

We thank for recommending our manuscript for publication and suggesting an interesting continuation of our work.